# Pharmacodynamic Profiling of Amoxicillin: Targeting Multidrug-Resistant Gram-Positive Pathogens *Staphylococcus aureus* and *Staphylococcus pseudintermedius* in Canine Clinical Isolates [note 1]

**DOI:** 10.3390/antibiotics14010099

**Published:** 2025-01-16

**Authors:** Syed Al Jawad Sayem, Ga-Yeong Lee, Muhammad Aleem Abbas, Seung-Chun Park, Seung-Jin Lee

**Affiliations:** 1Laboratory of Veterinary Pharmacokinetics and Pharmacodynamics, Institute for Veterinary Biomedical Science, College of Veterinary Medicine, Kyungpook National University, Daegu 41566, Republic of Korea; aljawadsayem@knu.ac.kr (S.A.J.S.); yeong1129@knu.ac.kr (G.-Y.L.); syedaleemabbas77@gmail.com (M.A.A.); 2Developmental and Reproductive Toxicology Research Group, Korea Institute of Toxicology, Daejeon 34114, Republic of Korea

**Keywords:** amoxicillin, *Staphylococcus aureus*, *Staphylococcus pseudintermedius*, minimal inhibitory concentration, time-kill assay, post-antibiotic effect

## Abstract

The rising threat of antimicrobial resistance (AMR) is a global concern in both human and veterinary medicine, with multidrug-resistant (MDR) pathogens such as *Staphylococcus aureus* and *Staphylococcus pseudintermedius* presenting significant challenges. **Background/Objectives**: This study evaluates the effectiveness of amoxicillin against these MDR pathogens in canine isolates using pharmacokinetic and pharmacodynamic parameters. **Methods**: Minimum inhibitory concentration (MIC), minimum bactericidal concentration (MBC), and mutation prevention concentration (MPC) were assessed. Additionally, time-kill assays and post-antibiotic effect (PAE) assessments were performed. Epidemiological cutoff (ECOFF) values were established for both species to guide therapy. **Results**: *S. aureus* had a higher resistance rate (35.89%) than *S. pseudintermedius* (15.27%), with MIC50 values of 0.50 μg/mL and 0.25 μg/mL, respectively. The MPC analysis revealed that *S. pseudintermedius* required higher antibiotic concentrations (16.11 μg/mL) to prevent mutations compared to *S. aureus* (2.20 μg/mL). Time-kill assays indicated that higher amoxicillin dosages caused faster bacterial reduction. The PAE analysis showed extended post-treatment bacterial suppression at elevated doses, particularly against *S. aureus*. **Conclusions**: Species-specific amoxicillin dosing strategies are necessary due to differing resistance and susceptibility profiles between *S. aureus* and *S. pseudintermedius*. High-dose amoxicillin therapy is recommended to achieve optimal therapeutic outcomes for resistant SA, while slightly adjusted dosing can manage *S. pseudintermedius* infections. These findings provide essential insights for veterinary antimicrobial stewardship, underscoring the need for tailored therapeutic approaches to minimize AMR development while ensuring effective infection control.

## 1. Introduction

The escalating global trend of antimicrobial resistance (AMR) poses a significant threat to both human and veterinary medicine. One of the primary concerns in veterinary medicine is the increasing prevalence of multidrug-resistant (MDR) pathogens such as *Staphylococcus aureus* and *Staphylococcus pseudintermedius* [1]. These Gram-positive bacteria are major culprits in various animal infections, particularly in canines, causing pyoderma, wound infections, and otitis [2,3]. The rising incidence of infections with resistant strains, including methicillin-resistant *S. aureus* (MRSA) and *S. pseudintermedius*, has reduced available treatment options, urging the need for more robust and targeted therapeutic strategies [4].

MRSA represents a major contributor to hospital-acquired infections, especially considering the rising prevalence of such infections within intensive care units in Japan [5,6]. Previously, MRSA detection rates increased across veterinary professionals and companion animals, with its prevalence among veterinary staff reaching 17.90% in the UK, 10% in Japan, and 3% in Denmark [7,8,9,10]. However, MRSP isolates were found in both hospitalized (46.20%) and outpatient dogs (19.40%) at a Japanese veterinary teaching hospital [9]. Furthermore, other methicillin-resistant coagulase-positive staphylococci have been reported in dogs and a veterinarian [11,12,13]. These pathogens carry *mecA*, which encodes penicillin-binding protein 2, reducing their affinity for β-lactam antibiotics [11].

Amoxicillin, a beta-lactam antibiotic, serves as a fundamental agent in the management of bacterial infections in humans and animals. Beta-lactams work by binding to penicillin-binding proteins, which inhibit the critical process in cell wall synthesis—transpeptidation. Furthermore, this binding activates autolytic enzymes within the bacterial cell wall [14]. These enzymes lyse the cell wall, thereby destroying the bacterial cell, which is referred to as bactericidal killing [15]. This amino-penicillin is created by adding a hydroxyl group at the para-position of its phenyl ring to counter antimicrobial resistance [16]. Amoxicillin covers various Gram-positive bacteria, with some added Gram-negative coverage compared to penicillin [17]. However, the effectiveness of amoxicillin is increasingly compromised due to the emergence of resistant strains. This underscores the importance of not only understanding the pharmacokinetics (PK) and pharmacodynamics (PD) of amoxicillin but also optimizing its use in combating MDR pathogens in veterinary settings.

PK describes how the body absorbs, distributes, metabolizes, and excretes a drug, whereas PD focuses on the drug’s effects on the organism, including its mechanism of action and antimicrobial activity. Integrating PK/PD data is essential to establish dosing regimens that maximize drug efficacy while minimizing the risk of resistance development [18].

Amoxicillin’s efficacy is primarily time-dependent; hence, maintaining drug concentrations above the minimum inhibitory concentration (T > MIC) for a sufficient duration is key to achieving therapeutic effects. However, the increasing resistance of *S. aureus* and *S. pseudintermedius* complicates the optimization of dosage regimens [19]. Alongside the growing resistance, the need for higher drug concentrations or prolonged exposure, which can cause toxicity and the development of further resistance, also increases. Therefore, establishing PD profiles that help clinicians tailor amoxicillin dosing strategies specifically against resistant pathogens is crucial to address the above-mentioned issue.

Determining the PD metrics provides a comprehensive understanding of an antibiotic’s potency and ability to prevent the emergence of resistant bacteria. This study aims to comprehensively assess the PD profiling of amoxicillin, focusing on its efficacy against multidrug-resistant Gram-positive pathogens, specifically, *S. aureus* and *S. pseudintermedius*, in canine clinical isolates. Therefore, this study aimed to (1) establish the MIC, MBC, MPC, time-kill curve, and post-antibiotic effect (PAE) of amoxicillin against *S. aureus* and *S. pseudintermedius* isolates in vitro. The MIC data were used to determine epidemiological cutoff values, as the PK/PD cutoff is an important measure for assessing antibiotic effectiveness in treating bacterial infections and is closely related to clinical results [20].

Additionally, the study aims to provide insights into optimal dosing strategies for amoxicillin to effectively target these resistant bacterial strains in veterinary medicine, facilitating future PK/PD-based dosing estimations to optimize its antibacterial activity against multidrug-resistant *S. aureus* and *S. pseudintermedius* in dogs. The findings will help clinicians make evidence-based decisions regarding amoxicillin use in veterinary practice, particularly for treating infections caused by resistant strains. This data-driven approach could also contribute to developing guidelines for optimal antibiotic use, ensuring both the efficacy and the prevention of further resistance development.

## 2. Results

### 2.1. Evaluation of MIC MBC and ECOFF in Antimicrobial Studies

This study assessed the antimicrobial activity of amoxicillin against *S. aureus* and *S. pseudintermedius* by determining the MIC and MBC (Table 1). A total of 35.89% of *S. aureus* isolates were resistant to amoxicillin in contrast to 15.27% of amoxicillin-resistant *S. pseudintermedius* isolates. The percentage of sensitive isolates was 64.10% for *S. aureus* and 69.44% for *S. pseudintermedius*. The MIC range for *S. aureus* was 0.25–128 μg/mL, while that for *S. pseudintermedius* was 0.125–64 μg/mL. The MBC range for *S. aureus* was 0.25–512 μg/mL, whereas that for *S. pseudintermedius* was 0.125–128 μg/mL. MIC_50_ values were 0.50 μg/mL for *S. aureus* and 0.25 μg/mL for *S. pseudintermedius*, while the MIC_90_ was 8 μg/mL for both species. Regarding bactericidal activity, the MBC_50_ value for *S. aureus* was 0.50 μg/mL compared to 1 μg/mL for *S. pseudintermedius*. The MBC_90_ was 8 μg/mL for both bacterial species. The MIC/MBC_50_ ratio for *S. aureus* was 1, indicating stronger bactericidal activity at lower concentrations, compared to *S. pseudintermedius*, which had a ratio of 0.25. Both species showed a MIC/MBC_90_ ratio of 1 (Table 2).

The EUCAST ECOFFinder analysis evaluated the ECOFF for *S. aureus* and *S. pseudintermedius*. For *S. aureus*, the rounded-up ECOFF value was 0.50 μg/mL for the 95.0%, 97.50%, 99%, 99.50%, and 99.90% thresholds. In contrast, *S. pseudintermedius* showed rounded-up ECOFF values of 0.50 μg/mL for 97.50% and 1 μg/mL for 99% thresholds. These findings highlight the defined thresholds for interpreting antimicrobial susceptibility for both species, which are crucial for guiding effective treatment strategies against infections caused by these pathogens (Figure 1).

### 2.2. Mutation Prevention Concentration Analysis

This study aimed to determine the MPC of amoxicillin against *S. aureus* and *S. pseudintermedius*. By plotting the concentration of the agent versus the log colony-forming unit (CFU)/mL, linear regression analysis was performed to estimate the drug concentration at which bacterial growth is entirely inhibited (Y = 0). For *S. aureus*, the linear regression equation was y = −5.86x + 12.89, and the extrapolated MPC value was 2.20 μg/mL. In the case of *S. pseudintermedius*, the regression equation was y=−0.71x+11.44, and the corresponding MPC value was 16.11 μg/mL. The data indicated that *S. pseudintermedius* requires a higher concentration (16.11 μg/mL) to achieve complete inhibition compared to *S. aureus* (Figure 2). These findings emphasize the variation in bacterial susceptibility to the antimicrobial agent. The higher MPC for *S. pseudintermedius* suggests that more potent dosing may be required to prevent resistance, particularly in clinical settings with high bacterial load.

### 2.3. Time-Kill Assay

The time-kill data of amoxicillin against *S. aureus and S. pseudintermedius*, as illustrated in the Figure 3, highlights the bactericidal effectiveness of various drug dosages (CT, 1/2M, 1M, 2M, and 4M) over 24 h. In both cases, the control treatment (CT) showed a continuous increase in bacterial concentration, suggesting reduced or delayed bactericidal action compared to other dosages. For *S. pseudintermedius*, higher dosages such as 1M, 2M, and 4M showed a rapid decrease in bacterial concentration within the first 2 h, approaching the detection limit, thus signifying potent bacterial killing activity. The 1/2M dosage, while also showing a decrease, maintained higher bacterial concentration over time, indicating slower bactericidal action compared to higher doses. Similarly, in the case of *S. aureus*, the higher dosages of 1M, 2M, and 4M demonstrated a swift reduction in bacterial concentration, which almost approached the detection limit within 2 h, while the 1/2M dosage showed a moderate decline over time. Overall, the data across both bacterial species indicated that higher amoxicillin dosages (1M, 2M, and 4M) are significantly more effective in rapidly reducing bacterial concentrations, as evidenced by their proximity to the detection limit in a time-dependent manner.

### 2.4. Post-Antibiotic Effect

The PAE of amoxicillin against *S. pseudintermedius* and *S. aureus* was assessed over 8 h (Figure 4). In both cases, the CT exhibited continuous bacterial growth, with *S. pseudintermedius* reaching approximately 12 log CFU/mL and *S. aureus* reaching 10 log CFU/mL by 8 h, indicating the absence of bactericidal activity. In contrast, the antibiotic-treated groups demonstrated significant differences in their PAEs depending on the dosage. For *S. pseudintermedius*, the highest PAE was observed in the 4M group. After 2 h of incubation, followed by the removal of the antibiotic, the bacterial count decreased to 4 log CFU/mL. However, the count gradually increased to 7.89 log CFU/mL by 8 h. The PAE duration for 4M was 3.40 h, while 1/2M had the shortest PAE duration of 1.40 h. Similarly, for *S. aureus*, the 4M group exhibited the most substantial PAE, with the bacterial load reducing to 3 log CFU/mL after 2 h and increasing slightly thereafter, with a PAE of 3.90 h. Additionally, lower dosages (such as 1M and 1/2M) demonstrated PAEs but with less pronounced reductions in bacterial load. Figure 5 highlights that higher amoxicillin concentrations (4 MIC) cause a more prolonged PAE for both pathogens, with *S. aureus* consistently having a longer PAE than *S. pseudintermedius* at each dosage. Therefore, amoxicillin is more effective at higher doses, especially against *S. aureus*.

## 3. Discussion

Antimicrobial resistance develops due to the adaptability of bacterial genes that constantly change to increase survival and drug resistance. Currently, >95% of staphylococcal isolates exhibit β-lactamase-mediated resistance. Additionally, MRSA accounts for 25–50% of clinical isolates in North America, Europe, and Asia [21]. A study analyzed the resistance profiles and clonal distribution of 103 methicillin-resistant *S. pseudintermedius* isolates from Europe and North America. The findings indicated that two main clonal lineages, ST71 and ST68, dominate these regions, showing significant resistance to key veterinary antibiotics, making MRSP infections a treatment challenge [22]. This study provides valuable insights into the antimicrobial effectiveness of amoxicillin against *S. aureus* and *S. pseudintermedius*, revealing significant differences in susceptibility, MIC, MBC, and other bactericidal metrics between the two bacterial species. These insights contribute to the broader understanding of antimicrobial resistance in common pathogens, with implications for clinical treatment.

The MIC and MBC analyses showed that *S. aureus* and *S. pseudintermedius* respond differently to amoxicillin, underscoring the variance in their antimicrobial resistance patterns. The MIC range observed for *S. aureus* (0.25–128 μg/mL) and *S. pseudintermedius* (0.125–64 μg/mL) suggests that *S. pseudintermedius* is generally more susceptible to amoxicillin, with a lower MIC50 value (0.25 μg/mL) compared to *S. aureus* (0.50 μg/mL). However, resistance rates are significantly higher in *S. aureus* (35.89%) than in *S. pseudintermedius* (15.27%), which is consistent with prior studies identifying *S. aureus* as a more resilient pathogen with diverse resistance mechanisms, including β-lactamase production and biofilm formation [23,24]

The MBC to MIC ratio is critical in assessing an antibiotic’s potential to kill bacterial cells rather than merely inhibiting their growth. A ratio of ≤4 is regarded as bactericidal, distinguishing bactericidal and bacteriostatic agents [25,26,27]. This threshold helps clinicians predict the likelihood of an antibiotic’s effectiveness in eradicating infections, especially when the immune response is compromised or limited. A ratio exceeding 4 typically suggests that the antibiotic inhibits bacterial growth without killing the organism, thus guiding appropriate therapeutic choices for bacterial infections. In our study, the bactericidal activity of amoxicillin is reflected in the MBC values. The MBC range for *S. aureus* (0.25–512 μg/mL) was wider than that of *S. pseudintermedius* (0.125–128 μg/mL); thus, higher concentrations of amoxicillin are required to eradicate *S. aureus*. The MIC/MBC_50_ ratio of 1 for *S. aureus* indicated a stronger bactericidal effect at lower concentrations, aligning with clinical findings that *S. aureus* often necessitates escalated doses for effective treatment. These findings are critical for clinicians, supporting higher amoxicillin doses to treat *S. aureus* infections, while lower concentrations may suffice for *S. pseudintermedius* (Table 2).

The European Committee on Antimicrobial Susceptibility Testing (EUCAST) developed the ECOFF to determine the MIC level indicating acquired resistance in bacterial isolates [28]. ECOFFs rely exclusively on bacterial phenotypic traits, even with advancements in genetic testing, and set the standard wild-type range for species regardless of their location or source [29]. The ECOFF for *S. aureus* was 0.5 μg/mL at several thresholds (95%, 97.50%, 99%, and 99.50%), while *S. pseudintermedius* displayed ECOFF values of 0.50 μg/mL at the 97.50% threshold and 1 μg/mL at the 99% threshold (Figure 1). These ECOFF values serve as valuable reference points for distinguishing between susceptible and resistant isolates. They further guide therapeutic decision making, facilitating more accurate dosing based on susceptibility patterns. This standardized approach to defining susceptibility through ECOFF thresholds offers the potential to minimize resistance development by promoting appropriate dosages in clinical practice. Given the differences in ECOFF values, *S. pseudintermedius* may require slightly higher doses in resistant cases, underscoring the importance of species-specific ECOFFs in guiding effective antimicrobial therapy.

The MPC is critical for understanding the dosage required to limit the emergence of resistant bacterial strains. The current study demonstrated a considerable difference in the MPC for a selective sensitive strain of *S. aureus* (2.20 μg/mL) and *S. pseudintermedius* (16.11 μg/mL), with *S. pseudintermedius* necessitating a higher concentration to achieve total inhibition (Figure 2). These values underscore a heightened susceptibility of *S. aureus* to amoxicillin, while *S. pseudintermedius* may develop resistance at lower concentrations, necessitating more potent dosing to preclude resistance development. The higher MPC for *S. pseudintermedius* implies that standard therapeutic concentrations may not entirely suppress mutation development in this species, especially in cases of high bacterial load. The differences in MPC values reflect the variability in resistance mechanisms, with *S. pseudintermedius* potentially possessing more robust defense mechanisms against mutation under antimicrobial pressure.

The time-kill assay results highlight the bactericidal effect of amoxicillin over 24 h, with pronounced reductions in bacterial counts at higher dosages (1M, 2M, and 4M MIC). Both *S. aureus* and *S. pseudintermedius* showed rapid declines in bacterial concentration at these elevated doses, underscoring the time-dependent efficacy of amoxicillin in achieving bactericidal effects. For *S. pseudintermedius*, concentrations approaching the detection limit were reached within 2 h, while *S. aureus* displayed similar bactericidal effects, albeit with slightly slower time kinetics. Therefore, the dosage increase not only augments bactericidal effects but also expedites bacterial eradication, an essential factor in clinical settings where rapid bacterial clearance can mitigate disease severity. Lower doses (1/2M) yielded slower kill rates, particularly for *S. pseudintermedius*. Hence, subtherapeutic dosing may result in prolonged bacterial survival and potentially increased resistance risks. Consequently, these results support the administration of higher amoxicillin doses in acute infections when rapid bactericidal activity is paramount.

Higher doses (4M MIC) yielded extended PAEs for both bacterial species, with *S. aureus* demonstrating a slightly longer PAE (3.90 h) than *S. pseudintermedius* (3.40 h). The PAE informs dosing intervals and helps reduce the risk of resistance development by ensuring prolonged bacterial suppression without continuous drug exposure (Figure 4 and Figure 5). We showed that higher amoxicillin doses produced more sustained PAEs, particularly in *S. aureus*, which could lead to fewer dosing requirements and lower treatment costs without compromising therapeutic outcomes. In the clinic, the extended PAE suggests that high-dose amoxicillin therapy may be particularly advantageous for *S. aureus* infections, in which longer PAE durations could reduce bacterial regrowth during dosing intervals.

The differential MIC, MBC, ECOFF, and MPC values between *S. aureus* and *S. pseudintermedius* have significant implications for optimizing amoxicillin therapy in clinical practice. For *S. aureus*, standard or slightly higher doses may suffice given the lower MPC and extended PAE, allowing for efficient bacterial eradication without excessive dosing and toxicity. In contrast, *S. pseudintermedius* infections may benefit from more aggressive dosing regimens, particularly in infections with a high bacterial load, to achieve complete bacterial eradication and limit resistance. The higher MPC for *S. pseudintermedius* underscores the importance of further research into resistance mechanisms in this species, particularly as it pertains to mutation rates under antimicrobial exposure. Future studies should focus on elucidating the genetic determinants underlying the observed differences in MPC and ECOFF values, as these could reveal novel targets for antimicrobial strategies or adjunct therapies to amoxicillin. However, this study is limited by the small and geographically confined sample of canine clinical isolates, which may restrict the generalizability of the findings. Additionally, resistance patterns evolve over time, underscoring the need for periodic validation and updates to ensure the relevance of the results.

## 4. Materials and Methods

### 4.1. Chemicals, Reagents, and Bacterial Strains

Amoxicillin powder was acquired from Sigma-Aldrich Co. (St. Louis, MO, USA). The bacterial growth medium, Mueller Hinton Broth (MHB), was obtained from Difco, Franklin Lakes, NJ, USA. Each reagent and chemical of analytical grade was employed in the current investigation.

### 4.2. S. aureus and S. pseudintermedius Strains, Culture Conditions, and Media

The Animal and Plant Quarantine Agency provided 78 strains of *S. aureus* and 72 strains of *S. pseudintermedius* from Kimcheon, Republic of Korea. Moreover, quality control strains for *S. aureus* and *S. pseudintermedius* were acquired from the American Type Culture Collection (ATCC) in Manassas, VA, USA. Each strain was grown in Mueller-Hinton broth (MHB) and subcultured thrice to attain stable growth in MHB. All *S. aureus* and *S. pseudintermedius* strains were grown in Mueller-Hinton Broth (Difco, USA) at 37 °C as shown in the Appendix A. The bacteria were kept overnight in broth at 37 °C in a shaking incubator before the experiments. Mueller-Hinton II (cation-adjusted) (Difco, USA) was utilized in all experiments including antibacterial drugs.

### 4.3. Minimum Inhibitory and Bactericidal Concentration

The MIC of amoxicillin was determined using the broth microdilution method following Clinical and Laboratory Standards Institute guidelines [30]. Two-fold serial dilutions of amoxicillin were performed in Mueller-Hinton II Broth (MHB II-Difco-BD, Franklin Lakes, NJ, USA) employing 96-well plates. The baseline concentration of amoxicillin for all cultures was 1024 μg/mL. The overnight culture was diluted to a concentration of 10^5^ CFU/mL of *S. aureus* and *S. pseudintermedius* strains, followed by allocation to the specified wells of the 96-well plates. The plates were incubated for 18–20 h at 37 °C, with results recorded using a microplate reader (Biotech EPOCH2, Winooski, VT, USA) at 600-nm wavelength. Twenty μL of the microtiter plate suspension from the MIC was inoculated onto MH agar to ascertain the MBC. The plates were incubated at 37 °C for 24 h to detect potentially slow-growing bacteria. The test was conducted in duplicate. The MIC findings were evaluated to ascertain the ECOFF values and processed using the Excel-based ECOFFinder tool (version 2.1). 

### 4.4. Mutation Prevention Concentration

This study aimed to determine the MPC of amoxicillin against *S. aureus* and *S. pseudintermedius.* By plotting the concentration of the agent versus the log colony-forming unit (CFU)/mL, linear regression analysis was performed to estimate the drug concentration at which bacterial growth is entirely inhibited (Y = 0). The MPC was determined following the methodology outlined in the previous study [31]. Briefly, *S. aureus* and *S. pseudintermedius* were cultured in MHB and incubated for 24 h. Afterwards, the suspension was centrifuged at 4000× *g* for 10 min and re-suspended in MHB to reach a concentration of 10^10^ CFU/mL. The inoculation was confirmed through serial dilution and plating of 100 μL samples on a drug-free medium. A series of agar plates, each containing amoxicillin, were inoculated with approximately 10^10^ CFU of the bacterial culture. Alongside testing the two-fold increases in MICs (2×, 4×, 8×, 16×, etc.), intermediate concentrations were evaluated to obtain more accurate MPC values. The plates were incubated at 37 °C for 48–72 h and checked visually for bacterial growth (Appendix A; Appendix A). The MPC was determined by plotting the logarithms of bacterial counts against amoxicillin concentrations, with the MPC identified where the plot intersected the *x*-axis. All experiments were performed in triplicate.

### 4.5. Time-Kill Assay

The time-kill assay was performed for the two selective intermediate strains as previously reported, with minor modifications [32]. *S. aureus* and *S. pseudintermedius* were inoculated into 5 mL of LB broth and incubated at 37 °C to obtain bacteria in the logarithmic growth phase. The final bacterial concentration was adjusted to 10^5^, and the organisms were exposed to amoxicillin at concentrations of 1/2×, 1×, or 4× MIC. The cultures were incubated at 37 °C and sampled at 0, 1, 2, 4, 6, 12, and 24 h. Then, the samples were serially diluted and plated on LB and TSB agar, followed by a 48 h incubation. After bacterial counting, the results were recorded.

### 4.6. Post-Antibiotic Effect

The PAE was assessed using previously established methods [33]. Selected strains were cultured in MH broth at 37 °C until reaching the logarithmic growth phase, resulting in a final inoculum of 1.5 × 10^10^ CFU/mL, which was then adjusted to 10^5^ CFU/mL for the experiment. The organisms were exposed to amoxicillin concentrations of 1/2×, 1×, or 4× MIC, while growth controls were simultaneously prepared without antibiotics. The tubes were placed in a 37 °C shaker for 2 h. Then, antibiotics were removed by a 1:1000 dilution with sterile broth. Controls underwent the same process. Following dilution, the tube contents were incubated at 37 °C until turbidity developed. Bacterial counts were conducted at 0, 1, 2, 3, 4, 5, and 6 h after the dilution. The PAE was calculated using the formula: PAE=T−C, where T is the time required for the bacterial count to increase by 1 log_10_ after dilution, and C is the time for the control without antibiotic exposure. All experiments were conducted in triplicate.

## 5. Conclusions

This study highlights the differential antimicrobial resistance profiles of *S. aureus* and *S. pseudintermedius* to amoxicillin, emphasizing the need for species-specific dosing strategies. *S. aureus* showed higher resistance (35.89%) compared to *S. pseudintermedius* (15.27%), with MIC ranges of 0.25–128 μg/mL and 0.125–64 μg/mL, respectively. The MBC range for *S. aureus* was broader than that for *S. pseudintermedius*, reflecting the need for higher doses to eradicate *S. pseudintermedius*. The MPC for *S. pseudintermedius* (16.11 µg/mL) was significantly higher than that for *S. aureus*, indicating a greater risk of resistance. These findings support tailored dosing regimens to improve outcomes and limit resistance development. Further research into resistance mechanisms and periodic updates of susceptibility thresholds are essential for refining treatment strategies and combating antimicrobial resistance in these pathogens.

## Figures and Tables

**Figure 1 antibiotics-14-00099-f001:**
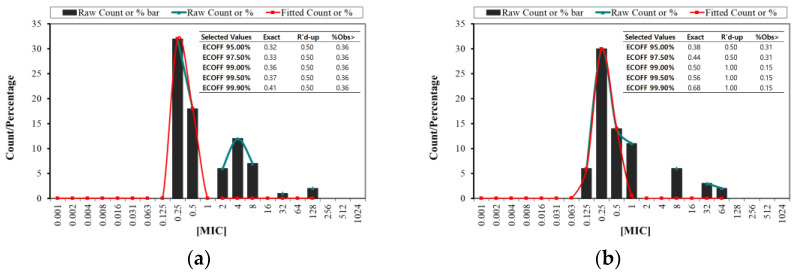
The fitted MIC distribution of amoxicillin against *S. aureus* (**a**) and *S. pseudintermedius* (**b**) was analyzed using the ECOFFinder.

**Figure 2 antibiotics-14-00099-f002:**
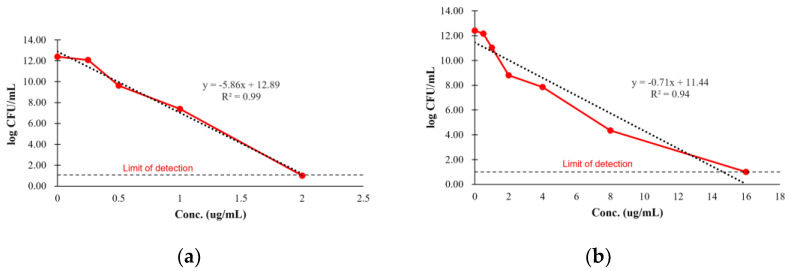
Mutation prevention concentration of *S. aureus* (**a**) and *S. pseudintermedius* (**b**) against amoxicillin. The limit of detection was considered as 1 log CFU/mL.

**Figure 3 antibiotics-14-00099-f003:**
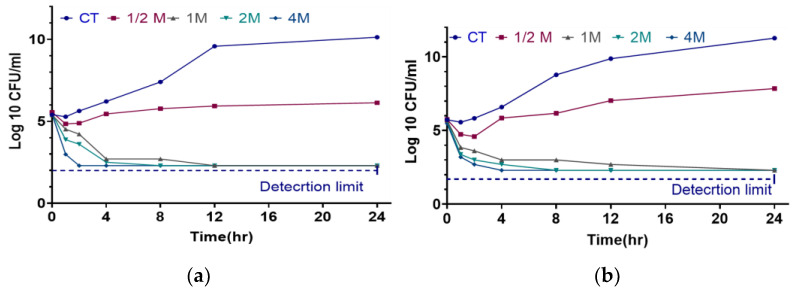
In vitro time-kill curves of amoxicillin against *S. aureus* (**a**) and *S. pseudintermedius* (**b**) at 0, 1, 2, 4, 8, 12, and 24 h containing 1/2×, 1×, 2×, and 4× MIC of amoxicillin and control (without drug) following 24 h incubation with 2-log CFU/mL detection limit.

**Figure 4 antibiotics-14-00099-f004:**
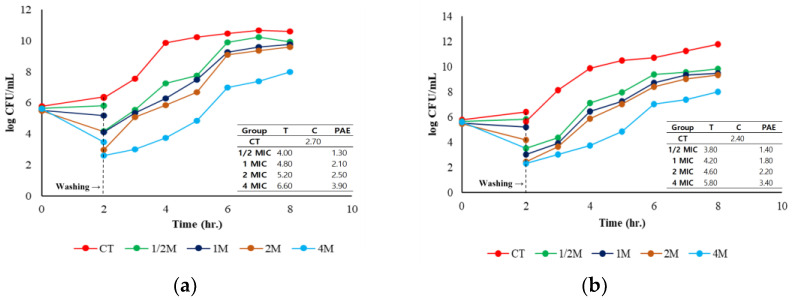
Post-antibiotic effect (PAE) following exposure to amoxicillin (1/2×, 1×, 2×, and 4× MIC) against *S. aureus* (**a**) and *S. pseudintermedius* (**b**).

**Figure 5 antibiotics-14-00099-f005:**
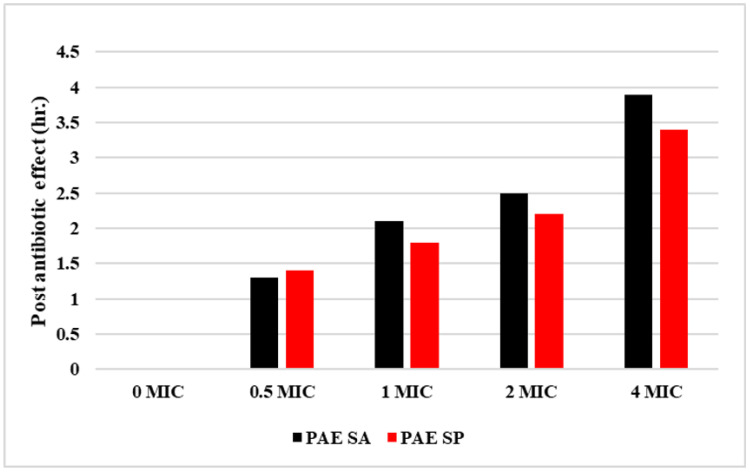
Comparative post-antibiotic effect of amoxicillin against *S. aureus* and *S. pseudintermedius*.

**Table 1 antibiotics-14-00099-t001:** MIC distribution of amoxicillin against *S. aureus* and *S. pseudintermedius*.

Species	S			I	R						
MIC (μg/mL)	**0.125**	**0.25**	**0.5**	**1**	**2**	**4**	**8**	**16**	**32**	**64**	**128**
SA		32	18		6	12	7		1		2
SP	6	30	14	11			6		3	2	

S, sensitive; I, intermediate; R, resistance; SA, *S. aureus*; SP, *S. pseudintermedius*.

**Table 2 antibiotics-14-00099-t002:** Comparative activity of amoxicillin against clinical *S. aureus* and *S. pseudintermedius* isolates from dogs.

	*S. aureus*	*S. pseudintermedius*
Observations	78	72
Distributions	7	7
% Resistance	35.89	15.27
% Sensitive	64.10	69.44
MIC range	0.25–128 μg/mL	0.125–64 μg/mL
MBC range	0.25–512 μg/mL	0.125–128 μg/mL
MIC_50_	0.50 μg/mL	0.25 μg/mL
MIC_90_	8	8
MBC_50_	0.50	1
MBC_90_	8	8
MIC/MBC_50_	1	0.25
MIC/MBC_90_	1	1

## Data Availability

Data are contained within the article.

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
