# Peer review of "Pharmacodynamic Profiling of Amoxicillin: Targeting Multidrug-Resistant Gram-Positive Pathogens Staphylococcus aureus and Staphylococcus pseudintermedius in Canine Clinical Isolates"

_antibiotics, 2025, doi:10.3390/antibiotics14010099_

Round 1

Reviewer 1 Report

Comments and Suggestions for Authors

1、Please carefully review all data throughout the text and ensure that all numerical values have been uniformly rounded according to the rule of rounding to the nearest integer or decimal place, as applicable, to maintain consistency and accuracy in the data presentation.

2、Please thoroughly examine and correct the unit expressions throughout the entire text, ensuring that the use of units such as "ug/mL" and "CFU/mL" conforms to scientific standards.

3、Explain the setup for selecting points on the regression curve in Figure 2(a) and whether the correlation between the regression curve and the actual data has any impact on the research findings.

4、The description in the results of the time-kill assay is notably inconsistent with Figure 3(b), stating that "For S. pseudintermedius, higher dosages such as 1M, 2M, and 4M showed a rapid decrease in bacterial concentration within the first 2 hours, approaching or falling below the detection limit" whereas the figure does not indicate any concentrations below the detection limit. Please verify and correct this description accordingly.

5、In the PAE experiment, the description "For S. pseudintermedius, the highest PAE was observed in the 4M group, with a reduction to 4 log CFU/mL after 2 hours and a gradual increase to 6 log CFU/mL at 8 hours" is ambiguous. It is unclear whether the "2 hours" refers to the time before or after dilution drug; if it is after dilution, it should be "4 hours". Additionally, it is recommended to add graduations to the vertical axis of the graph, as the bacterial count at 8 hours for the 4M concentration clearly exceeds 6 log CFU/mL.

6、In the "Mutation Prevention Concentration Analysis," please explain the rationale behind adding fluoroquinolones to the plates.

7、In the Supplementary Materials, please check if the labeling of the dilution factors in "No Treatment" of Figure 2 is correct.

Comments on the Quality of English Language

The English could be improved.

Author Response

Comment 1: Please carefully review all data throughout the text and ensure that all numerical values have been uniformly rounded according to the rule of rounding to the nearest integer or decimal place, as applicable, to maintain consistency and accuracy in the data presentation.

Response 1: Thank you for highlighting this. We fully agree with the comment and have taken steps to ensure that all numerical data are consistently rounded and corrected to maintain uniformity in integer and decimal placement.

Comment 2: Please thoroughly examine and correct the unit expressions throughout the entire text, ensuring that the use of units such as "ug/mL" and "CFU/mL" conforms to scientific standards.

Response 2: We agree and have updated the units throughout the text to align with scientific standards.

Comment 3: Explain the setup for selecting points on the regression curve in Figure 2(a) and whether the correlation between the regression curve and the actual data has any impact on the research findings.

Response 3: Yes, the regression curve impacted the actual data, altering the research findings. Therefore, we corrected this figure and revised the results, highlighted in red color. Additionally, we excluded the growth curve beyond 2 µg/mL in Figure 2(a), as no changes were observed in the growth curve.

Comment 4: The description in the results of the time-kill assay is notably inconsistent with Figure 3(b), stating that "For S. pseudintermedius, higher dosages such as 1M, 2M, and 4M showed a rapid decrease in bacterial concentration within the first 2 hours, approaching or falling below the detection limit" whereas the figure does not indicate any concentrations below the detection limit. Please verify and correct this description accordingly.

Response 4: We agree as well as corrected this statement followed by clarifying it in the results section marked in red color.

Comment 5: In the PAE experiment, the description "For S. pseudintermedius, the highest PAE was observed in the 4M group, with a reduction to 4 log CFU/mL after 2 hours and a gradual increase to 6 log CFU/mL at 8 hours" is ambiguous. It is unclear whether the "2 hours" refers to the time before or after dilution drug; if it is after dilution, it should be "4 hours". Additionally, it is recommended to add graduations to the vertical axis of the graph, as the bacterial count at 8 hours for the 4M concentration clearly exceeds 6 log CFU/mL.

Response 5: Agree. We have corrected this statement and provided a clear explanation in both the results and the figure. The "2 hours" refers to the time before dilution, specifically the period between the start of incubation and the washing of the antibiotic (dilution). Additionally, the bacterial count at 8 hours for the 4M concentration was 7.89 log CFU/mL, added in the manuscript.

Comment 6: In the "Mutation Prevention Concentration Analysis," please explain the rationale behind adding fluoroquinolones to the plates.

Response 6: The description has been corrected to reflect that it was amoxicillin, not fluoroquinolones.

Comment 7: In the Supplementary Materials, please check if the labeling of the dilution factors in "No Treatment" of Figure 2 is correct.

Response 7: The correction has been made in supplementary materials.

Reviewer 2 Report

Comments and Suggestions for Authors

Introduction 

1. Introduction is well written but I suggest to cite the appropriate references in line 69-75. 

2. Line 90-94 may not be appropriate for the introduction section. Please rewrite in a better way. 

Methodology

3. Line 319 should be written in a better way. "The MPC was determined as described elsewhere..." This doe not looks good for a scientific paper, please write in a better and scientific way. 

4. In introduction, you talked about PK/PD data but you did not performed PK/PD modeling for dosing optimization, and paper lacks advanced PK/PD integration. Introducing such models could enhance the clinical relevance of the findings. This extends up to results and discussion. 

Results

1. Results section is fine there are numerous typos like in Figure 3, it is written as "Detecrtion" in stead of detection. In figure 2 it should be limit not "limite"

2. Line 133-136 is not part of result, it is part of methodology, similarly follow for other section also. 

Discussion

1. Line 208. What stands out your study from previous similar studies? 

2. Discussion is written well. But please add the limitation of the study and future research direction in las paragraph of the discussion. 

Conclusion

Real conclusion of the study ? Please rewrite.  

Author Response

Introduction 

Comment 1: Introduction is well written but I suggest to cite the appropriate references in line 69-75.

Response 1: Thank you for your suggestion. We agree. An appropriate reference has been added.

Comment 2: Line 90-94 may not be appropriate for the introduction section. Please rewrite in a better way.

Response 2: The statement was rewritten to reflect the objective of this study.

Methodology

Comment 3: Line 319 should be written in a better way. "The MPC was determined as described elsewhere..." This doe not looks good for a scientific paper, please write in a better and scientific way.

Response 3: The line has been rewritten scientifically and marked as red color.

Comment 4: In introduction, you talked about PK/PD data but you did not performed PK/PD modeling for dosing optimization, and paper lacks advanced PK/PD integration. Introducing such models could enhance the clinical relevance of the findings. This extends up to results and discussion.

Response 4: Thank you for your insightful comment. We have mentioned the PK/PD model in the introduction as a potential future direction for this study. Our current research provides the necessary data to support developing and applying PK/PD models for dosing optimization in subsequent studies.

Results

Comment 1: Results section is fine there are numerous typos like in Figure 3, it is written as "Detecrtion" in stead of detection. In figure 2 it should be limit not "limite"

Response 1: The correction has been made in the manuscript.

Comment 2: Line 133-136 is not part of result, it is part of methodology, similarly follow for other section also.

Response 2: We agree. This section was excluded from the result part and included in the methodology part marked as red color.

Discussion

Comment 1: Line 208. What stands out your study from previous similar studies?

Response 1: The reference to the previous study was added correctly.

Comment 2: Discussion is written well. But please add the limitation of the study and future research direction in las paragraph of the discussion.

Response 2: Thank you for these valuable suggestions, we have added the limitation of the study and future research direction in the last paragraph of the discussion.

Conclusion

Comment 1: Real conclusion of the study? Please rewrite.

Response 1: We have rewritten the conclusion.

Round 2

Reviewer 1 Report

Comments and Suggestions for Authors

1.The content in materials and methods is not indented first

2.Please review and adjust the unit to "μg/mL".

Comments on the Quality of English Language

The language level of the article needs to be improved.

Author Response

Comment 1: The content in Materials and Methods is not indented first.
Response 1: We sincerely thank the reviewer for pointing this out. The formatting issue in the Materials and Methods section has been carefully corrected, as suggested.

Comment 2: Please review and adjust the unit to "μg/mL".
Response 2: We appreciate the reviewer’s observation. The unit has been reviewed and appropriately adjusted to "μg/mL".